# Comparative Analysis of Volatiles Emitted from Tomato and Pepper Plants in Response to Infection by Two Whitefly-Transmitted Persistent Viruses

**DOI:** 10.3390/insects13090840

**Published:** 2022-09-15

**Authors:** Saptarshi Ghosh, Shoshana Didi-Cohen, Alon Cna’ani, Svetlana Kontsedalov, Galina Lebedev, Vered Tzin, Murad Ghanim

**Affiliations:** 1Department of Entomology, ARO, The Volcani Center, HaMaccabim Road 68, P.O. Box 15159, Rishon LeZion 7505101, Israel; 2Department of Entomology, University of Georgia, Griffin, GA 30223, USA; 3French Associates Institute for Agriculture and Biotechnology of Drylands, Jacob Blaustein Institutes for Desert Research, Ben-Gurion University of the Negev, Sede Boqer Campus, Be’er Sheva 8499000, Israel; 4Department of Food Sciences, University of Copenhagen, DK-1165 Copenhagen, Denmark

**Keywords:** *Bemisia tabaci*, volatiles, begomoviruses, poleroviruses, tomato, pepper

## Abstract

**Simple Summary:**

Viruses transmitted by the whitefly *Bemisia tabaci*, a cosmopolitan pest that inflicts damage on many agricultural crops, are among the most important plant viruses and cause devastating damage to agriculture worldwide. Begomoviruses and a new polerovirus transmitted by the whitefly cause serious damage to tomato and pepper crops in Israel, respectively. Here, both viruses were studied with regard to their ability to induce the production of volatile organic compounds (VOCs) upon plant infection, which are thought to mediate communication with the environment. The results showed that infection with each virus alone, or infestation with whiteflies, without viruses, induced shared and unique VOC accumulation. The VOCs identified in these unique responses suggest that plants can respond specifically to virus infection or insect infestation, and those VOCs can be used individually or as blends for monitoring and disrupting pest populations.

**Abstract:**

The whitefly *Bemisia tabaci* is one of the most important agricultural pests due to its extreme invasiveness, insecticide resistance, and ability to transmit hundreds of plant viruses. Among these, Begomoviruses and recombinant whitefly-borne Poleroviruses are transmitted persistently. Several studies have shown that upon infection, plant viruses manipulate plant-emitted volatile organic compounds (VOCs), which have important roles in communication with insects. In this study, we profiled and compared the VOCs emitted by tomato and pepper plant leaves after infection with the Tomato yellow leaf curl virus (TYLCV) (Bogomoviruses) and the newly discovered Pepper whitefly-borne vein yellows virus (PeWBVYV) (Poleroviruses), respectively. The results identified shared emitted VOCs but also uncovered unique VOC signatures for each virus and for whitefly infestation (i.e., without virus infection) independently. The results suggest that plants have general defense responses; however, they are also able to respond individually to infection with specific viruses or infestation with an insect pest. The results are important to enhance our understanding of virus- and insect vector-induced alteration in the emission of plant VOCs. These volatiles can eventually be used for the management of virus diseases/insect vectors by either monitoring or disrupting insect–plant interactions.

## 1. Introduction

The status of the whitefly *Bemisia tabaci* (*B. tabaci*; silverleaf whitefly) as one of the most important insect pests of crops has been unmatched due to its high invasiveness, insecticide resistance, and as supervectors of over three hundred diverse plant viruses [1,2]. *Bemisia tabaci* species complex members are transmitters of ssDNA viruses belonging to the genera Begomoviruses, in a circulative and persistent mode by the whitefly [1]. The Tomato yellow leaf curl virus (TYLCV) complex is one of the most devastating Begomovirus groups constraining global tomato (*Solanum lycopersicum*) cultivation, and it remains the most widely studied whitefly transmitted viruses [3,4]. In the last 20 years, emerging plant virus diseases transmitted by *B. tabaci* have heavily constrained crop production and the trade of agricultural commodities [5,6]. Moreover, the discovery of new associations of *B. tabaci* with virus genera that were previously not reported to be whitefly transmitted has magnified its threat to crop production [7,8].

Plants emit diverse volatile organic compounds (VOCs) constitutively into the surrounding atmosphere to communicate with plants and organisms at diverse trophic levels [9]. VOCs are the products of diverse biosynthetic pathways and are emitted as a mixture of molecules belonging to a wide range of biochemical classes, mainly phenylpropanoids, benzenoids, terpenoids, and aliphatics [9]. Apart from the role of VOCs in plant–plant and plant–pollinator communication, they play a key role in direct and indirect defense against biotic stresses [9,10]. The complex VOC blends released from plants into the atmosphere can mediate both direct defense by repelling herbivores and preventing oviposition [11], and indirect defense by attracting natural enemies such as parasitoids and predators [12]. Volatile molecules can be emitted either continuously, or upon demand as herbivore-induced plant volatiles, and are perceived by herbivores via olfactory sensing [13].

Phytohormones are involved in the response to *B. tabaci* infestation, and the most important ones include jasmonic acid (JA) and salicylic acid (SA) [14,15]. The two active forms of these phytohormones are methyl jasmonate (MeJA) and methyl salicylate (MeSA). MeJA is an integral component of the plant’s defense responses to insect feeding and mechanical damage to leaves [16,17], while MeSA is more dominant upon phloem feeding by insects [18]. Direct damage to plant tissue with the subtle feeding mechanism of *Bemisia tabaci* is relatively lesser than the damage induced by virus transmission, but is sufficient to induce MeSA and terpenoid emissions [19,20,21]. VOCs emitted by healthy plants differ from those infected by persistently transmitted viruses, which often modify the behavior of their insect vectors, resulting in increased spread [22]. Plants infected with persistently transmitted viruses often emit VOCs that attract nonviruliferous insect vectors to facilitate virus acquisition, whereas viruliferous insects often prefer non-infected plants to facilitate virus spread [23,24].

The VOC emission profiles of plants infected with persistently transmitted viruses belonging to the genus Begomovirus [25,26,27] or Luteoviridae [21,28,29,30,31] have been extensively studied to understand their effect on virus spread by insect vectors. The genetic material, virus structure, and the insect vector of Begomoviruses (ssDNA, geminate, whitefly) is much different from that of Luteoviridae (ssRNA, icosahedral, aphid). However, the circulative route and transmission mechanisms for both virus groups within their respective insect vectors have striking similarities. Plants infected with Begomoviruses or Luteoviruses have altered emissions of volatiles, which influence the behavior and performance of their insect vectors. The identification of unique and shared VOC changes induced by these two virus groups could clarify how the viruses attract their specific vectors. However, the alterations in VOCs induced solely by these two groups of viruses could not be compared due to their transmissibility by different vectors. A recombinant polerovirus (Luteoviridae), Pepper whitefly-borne vein yellows virus (PeWBVYV), with a high genetic identity to other poleroviruses known to infect pepper, was recently discovered to be exclusively transmitted by the whitefly in a circulative mode [7,32]. The transmission of PeWBVYV by whiteflies has been hypothesized to be an evolutionary strategy to rescue and maintain PeWBVYV from competing with aphid-transmitted poleroviruses [33]. The host preference (infected/non-infected) of whiteflies (non-viruliferous/viruliferous) for PeWBVYV remains to be determined; however, it is possible that PeWBVYV plants emit specific VOCs to attract whiteflies. Moreover, PeWBVYV infections of pepper plants manifest prominent symptoms under laboratory conditions, unlike the aphid-transmitted Pepper vein yellows virus 2 (PeVYV-2). In this study, we profiled and compared the VOCs emitted by tomato and pepper (*Capsicum annuum*) leaves after infection with PeWBVYV and TYLCV, respectively, with the objective of identifying common and unique VOCs induced by persistently transmitted whitefly-borne viruses. The outcome of this study would enhance our understanding of virus- and insect vector-induced alterations in the emission of plant volatiles and to use them for the management of the virus diseases/insect vectors.

## 2. Materials and Methods

### 2.1. Rearing and Maintenance of Non-Viruliferous and Viruliferous (TYLCV/PeWBVYV) Bemisia tabaci

Non-viruliferous *B. tabaci* were reared separately in insect-proof cages on virus uninfected tomato plants (*Solanum lycopersicum* cv. Avigail) and bell pepper plants (*Capsicum annuum* cv. Canon), under controlled conditions of 25 ± 5 °C, 60% R.H and a 14:10 hours diurnal period. Similarly, viruliferous *B. tabaci* were reared separately on Tomato yellow leaf curl virus (TYLCV)-infected tomato plants and Pepper whitefly-borne vein yellows virus (PeWBVYV)-infected peppers of the same varieties (Figure 1). The viruliferous/non-viruliferous status of the plants and whitefly adults was confirmed by PCR [7,34].

### 2.2. Inoculation of Plants with Viruliferous (TYLCV/PeWBVYV) Whiteflies and Viral Detection

TYLCV-viruliferous adults of MEAM1 species of *B. tabaci* (0–12 days of age) reared on TYLCV-infected tomato plants were collected and allowed a life-long inoculation access period on five uninfected tomato plants (3- to 4-leaf stage, 50 whiteflies/plant), inside insect-proof cages. Non-viruliferous whitefly adults of similar age and reared on virus uninfected tomato plants were released for life-long onto five uninfected tomato plants. A control with five uninfected tomato plants without whitefly inoculation was set up in parallel. TYLCV infections were confirmed in all five tomato plants inoculated with viruliferous *B. tabaci* three weeks after inoculation by PCR. Similarly, the uninfected status of the plants inoculated with non-viruliferous whitefly adults and the control plants was confirmed by PCR [34].

PeWBVYV non-viruliferous/viruliferous adults of MEAM1 species of *B. tabaci* (0–12 days of age) reared on uninfected or PeWBVYV-infected bell pepper plants were allowed a life-long inoculation access period onto five uninfected bell pepper plants (3-leaf stage, 50 whiteflies/plant) separately inside insect-proof cages. A control experiment with five pepper plants without whitefly inoculation was set up. The PeWBVYV infection status of pepper plants in all treatments was confirmed three weeks after inoculation by reverse transcription-PCR [7].

### 2.3. Volatile Extraction and Analysis Using Headspace Solid Phase Micro-Extraction (HS-SPME) Coupled with Gas Chromatography–Mass Spectrometry (GC-MS)

Three leaves (3rd, 4th, and 5th leaf from the top) were collected and pooled from each of the whitefly-inoculated or non-inoculated control plants (five biological repeats), three weeks after infestation, snap frozen in liquid nitrogen, and stored at −80 °C. VOCs were extracted and analyzed, as previously described [35,36]. In brief, leaf tissues were ground to a fine powder in liquid nitrogen and placed in 20 ml glass vials containing 1 g NaCl, and 7 ml NaCl solution (20% *w*/*v*). As an internal standard, 10 ppm of isobutylbenzene (Sigma-Aldrich, Rehovot, Israel) was added. The glass vials were incubated for 15 min at 60 °C in a PAL COMBI-xt autosampler (CTC Analytics AG Switzerland) to release free volatiles into the headspace. For volatile desorption, a 10 mm long solid phase micro-extraction (SPME) fiber was used and run in splitless mode at the inlet of a GC (Agilent 7890A, USA), coupled with an MS detector (Agilent 5977B, Santa Clara, CA, USA). The retention index (RI) was calculated by running a C_8_–C_20_ *n*-alkane mix (Sigma-Aldrich, Rehovot, Israel). Compounds were identified by Wiley 10 with NIST 2014 mass spectral library data using the MassHunter software package (version B.08.00, Agilent, USA). Validation was based on a comparison of mass spectra and retention indices. Where available, compounds were identified using authentic standards (Sigma-Aldrich, Rehovot, Israel), and analyzed under similar conditions.

### 2.4. Statistics

Data for one-way Analysis of Variance (ANOVA) and Student’s *t*-test were normalized by log-transformation and auto-scaling using the MetaboAnalyst 5.0 tool [37,38], *p* values were corrected using the false discovery rate (FDR), and compounds with an adjusted *p* value < 0.05 were retained. Venn diagrams were created using Venny2.1 (bioinfogp.cnb.csic.es/tools/venny, accessed on 15 January 2021). Bar graphs were created using JMP13 software (SAS; www.jmp.com, accessed on 1 August 2022) and Microsoft Excel.

## 3. Results

### 3.1. VOC Profiles of Tomato Plants Uninfected or Infected with TYLCV

The volatile content of tomato leaves of virus (TYLCV)-infected plants generated by inoculation with viruliferous whiteflies, non-infected tomato plants generated by inoculation with non-viruliferous whiteflies (NV), and non-inoculated control (NI) was determined using the HS-SPME-GC-MS technique. A total of 76 volatile compounds were detected and divided into classes: 26 fatty acid (FA) derivatives, 14 phenylpropanoids and benzenoids, 13 monoterpenes, 13 sesquiterpenes, 6 irregular terpenes, and 4 furans (Appendix A). A total of 16 VOCs of different classes were consistently altered in plants inoculated with whiteflies (TYLCV/NV) compared to the non-inoculated (NI) control plants, out of which three were common between the TYLCV and NV treatments (Figure 2A). VOCs profile of TYLCV-infected plants generated by inoculation with viruliferous whiteflies were mostly different than the NV (Figure 2A). However, the means of only 11 of these VOCs differed significantly upon one-way ANOVA analysis (Figure 2B). The VOC class of FA derivatives, including 3-methyl-2-heptanone and nonanoic acid, was significantly reduced upon whitefly inoculation (TYLCV, NV), while 1-octanol and 9-hexadecenoic acid and methyl ester showed opposite patterns (Figure 2A,B). Ethyl myristate levels were significantly lower in TYLCV-infected plants, whereas levels of 7,10,13-hexadecatrienoic acid and methyl ester were significantly increased in plants of the NV treatment (Figure 2B). VOCs belonging to the class of phenylpropanoids and benzenoids, such as guaiacol, 1,3-dimethylbenzene, and 1,2-dimethoxybenzene, were significantly higher in TYLCV infected tomato plants, while phenol was higher in tomato plants inoculated with whitefly (TYLCV, NV) than in the non-inoculated control (Figure 2B). Only one compound from the terpene class, 1,3,8-p-Menthatriene, was reduced in TYLCV-infected plants compared to NI. Overall, this analysis suggests that the VOC profiles of tomato leaves exposed to viruliferous and non-viruliferous whiteflies are mostly unique, and that the major changes in volatiles are associated with virus infection of plants.

### 3.2. VOC Profiles of Pepper Leaves Uninfected or Infected with PeWBVYV

A total of 93 volatile compounds were detected and divided into classes: 67 FA derivatives and GLVs, 15 phenylpropanoids and benzenoids, 2 monoterpenes, 5 irregular terpenes, and 4 furans (Appendix A) across all experimental treatments. Quantities of a total of 12 VOCs differed consistently in plants inoculated with non-viruliferous (NV) and viruliferous (PeWBVYV) whiteflies compared to non-inoculated plants (NI). Guaiacol levels increased under both PeWBVYV and NV treatments (Figure 3A). Notably, the phytohormone methyl salicylate (MeSA), belonging to the phenylpropanoids and benzenoids classes, was significantly increased in plants inoculated with non-viruliferous whitefly adults (NV) compared to the control (NI). In contrast to the previous experiment with TYLCV, in this experiment, the differing VOCs were predominantly detected in plants inoculated with non-viruliferous whitefly adults. The mean quantities of eight VOCs were significantly different between the NI control and NV plants using one-way ANOVA (Figure 3B). Synthesis of many other VOCs, such as methyl pentadecanoate, benzaldehyde, dihydroaplotaxene, methyl stearate, pentadecanal, and linalool, were significantly increased in NV plants. This analysis indicated that alterations in VOC biosynthesis in plants were primarily in response to whiteflies rather than the polerovirus infection.

Altered VOC profiles between the tomato and pepper plants inoculated with either nonviruliferous or viruliferous whiteflies were mostly unique compared to the non-inoculated control plants (Figure 4). This indicates a differential response between the two plant hosts to the two viruses. Only guaiacol and methyl palmitoleate were common VOC with increased quantities in response to whitefly infestation in tomato and pepper.

## 4. Discussion

Multiple species of viruliferous/non-viruliferous *B. tabaci*, upon interaction with diverse plants, induce changes in the emission of volatile compounds that either repel or attract whiteflies [25,26,27,39,40]. Differential visual and olfactory cues from virus-infected or uninfected plants orient insect vectors to facilitate virus spread. Plants infected with TYLCV attract non-viruliferous whiteflies, whereas viruliferous whiteflies prefer non-infected plants [26,41,42,43,44]. However, in response to whitefly infestation, volatiles vary between host plants [45,46] as well as the species of whitefly [47,48]. Herein, we compared the alteration in volatile response in tomato and pepper plants in response to whitefly infestation and post-infection with two diverse persistently transmitted viruses with the objective of identifying unique and shared patterns in volatile emission.

Our results indicated that TYLCV-infected plants had higher alterations in volatile emissions than non-inoculated plants (Figure 2). Similar changes in volatile profiles when comparing TYLCV-infected and uninfected control plants has been previously reported [49]. Similar to our findings, they report increase of VOCs belonging to the phenylpropanoid and benzenoid classes upon virus infection, while the VOCs belonging to the monoterpenoid class of compounds were reduced in TYLCV-infected plants. A recent study on rice (*Oryza sativa*) plants measured the VOCs upon infection with the *Rice dwarf virus* (RDV) and correlated with host preference of its insect vector, the green rice leafhoppers (*Nephotettix cincticeps*; Hemiptera: Cicadellidae). The volatile profile revealed that RDV infection significantly induced the emission of sesquiterpene and (E)-β-caryophyllene while other monoterpenes were not affected. Moreover, increased levels of the FA derivates and green leaf volatiles (GLVs), 2-heptanol in RDV-infected plants selectively influenced the olfactory behavior of the non-viruliferous and viruliferous insect vectors [50]. In this study, TYLCV infection of tomato plants led to increased emission of FA derivates and GLVs, such as 1-octanol and methyl palmitoleate, whereas quantities of ethyl myristate, nonanoic acid, and 3-methyl-2-heptanone were reduced. Alteration in the abundance of compounds belonging to this class of VOCs upon healthy whitefly infestation was previously studied in different plant species, including the common bean [51], tomato, tobacco, cabbage, cotton, cucumber, celery [39,47], and eggplant [52]. However, the change in their biosynthesis in response to TYLCV has been reported for the first time in this study. The roles of GLVs vary and might function as attractants to whiteflies [39] or their parasitoids [51], while they can be used to prime plant host defenses and reduce virus transmission [53]. Terpenoids are known repellents of whiteflies [25,54,55], with reduced accumulation in virus-infected plants compared to non-infected plants, facilitating the acquisition of plant viruses [26,27,41,42]. Future validation of the identified VOC changes with respect to the preference of viruliferous or non-viruliferous whiteflies will be helpful in designing effective management or monitoring strategies, such as designing new traps for monitoring, and lure and kill strategies against whiteflies and for reducing TYLCV transmission.

Previous studies have shown that infection of plants with poleroviruses significantly alters volatile emissions to arrest the aphid population on infected leaves and facilitate virus acquisition [21,29,31,56]. However, in this study, plants infected with PeWBVYV had minimal change in volatile emissions compared to plants inoculated with non-viruliferous whitefly adults. Interestingly, inoculation of pepper plants inoculated with non-viruliferous whiteflies had a more profound effect on the biosynthesis of volatiles than PeWBVYV-infected plants (Figure 3). Our previous study showed that PeWBVYV competes with another sympatric aphid-transmitted pepper-infecting polerovirus named Pepper vein yellows virus 2 (PeVYV-2) when co-infecting the host plant or insect vectors [33]. Unlike PeVYV-2, pepper plants maintained under laboratory conditions and infected with PeWBVYV display prominent symptoms of interveinal chlorosis and leaf yellowing. Thus, we expected major changes in the VOC profiles of PeWBVYV-infected plants for the whiteflies to locate diseased plants and avoid co-infections. The reasons for the insignificant effect of PeWBVYV on volatile emissions remain unknown. It is possible that the recombinant PeWBVYV still adapts to the host plant and induces notable physiological changes in the plant. Although PeWBVYV infections of plants had minimal olfactory changes to attract whitefly adults, it is still possible that they would be less attractive to aphids in the presence of plants infected with aphid-transmitted poleroviruses, and vice versa. However, this needs to be further investigated with a choice test between viruliferous and non-viruliferous whiteflies/aphids in the presence of the two poleroviruses. Our previous study showed that PeWBVYV has a lower transmission efficiency and a longer latent period inside the whiteflies compared to the aphid-transmitted poleroviruses (32). We speculate that the relationship between poleroviruses and whiteflies is still evolving and can be a factor in the observed lack of VOC changes in infected plants. Nevertheless, the VOC alterations seen in pepper plants infested with non-viruliferous whiteflies were mostly unique to that condition. Notably, one of the two compounds affected by whitefly infestation in the two plant species is guaiacol (Figure 4). A previous study on tomato leaves examined the effects of TYLCV infection hosted by whiteflies on peroxidase (POD) activity. The researchers used guaiacol as a substrate that stimulates POD activity in the presence of TYLCV [57]. This suggests that peroxidases, which are known to play a role in defense against infections [57,58], might also affect the two pathosystems studied here; however, it requires further validation.

The HS-SPME method, which evaluates VOC profiles synthesized in leaves, was used in this study. The dynamic headspace method can also be used to collect the volatiles emitted into the atmosphere [21,31]. While both methods are commonly used, HS-SPME provides a snapshot of the VOC status, whereas the dynamic headspace technique allows the collection of VOCs over a period of time [21,31,59] and the two methods often yield different results [59]. The VOC profile and intensity can also be affected by abiotic conditions. Thus, a follow-up experiment under field conditions for these crops should be conducted involving collection of VOCs using the dynamic headspace technique.

## 5. Conclusions

The identification of VOCs in this study shows that plants infested with viruliferous and non-viruliferous insects or plants infected with different whitefly-transmitted viruses share common emitted compounds, which might hint at the shared pathways that respond to insect infestation or infection with persistent phloem-restricted viruses. These pathways were reported for viruses belonging to Begomoviruses/Luteoviruses genera, such as those studied in the current project. However, the results also suggest that specific plants produce unique responses in relation to a specific virus infection or inoculation with the insect vector. These unique signatures, exemplified in the unique VOCs identified after each virus infection or insect infestation, can be used to design methods for the management of these virus–vector complexes and reduce virus transmission. Such methods include trapping and lure and kill, as well as breeding plants for less attractive and higher resistance to whiteflies.

## Figures and Tables

**Figure 1 insects-13-00840-f001:**
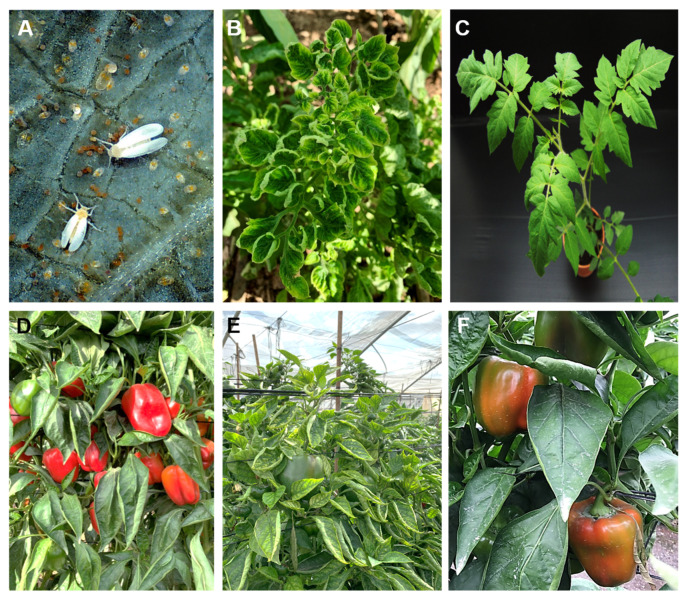
The two pathosystems compared in this study. The whitefly *Bemisia tabaci* (**A**) is the vector for the Tomato yellow leaf curl virus (TYLCV), which shows typical symptoms of yellowing and leaf curling (**B**) compared to a healthy plant (**C**). The whitefly is also a vector for Pepper whitefly-borne vein yellows virus (PeWBVYV), which shows typical symptoms of leaf curling and yellowing (**E**) and fruit discoloration (**F**) compared to uninfected plants (**D**).

**Figure 2 insects-13-00840-f002:**
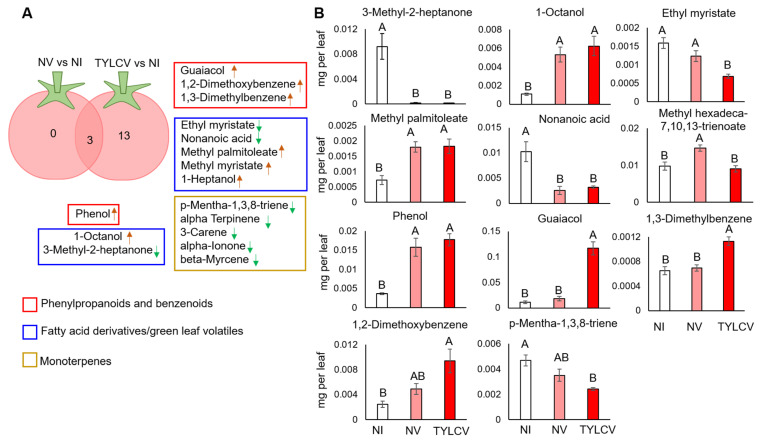
Altered VOC profiles of tomato plants. (**A**) Venn diagram representing the altered VOC profiles of tomato plants inoculated with viruliferous (TYLCV) and non-viruliferous whiteflies (NV) compared to non-inoculated control plants (NI) using Student’s *t*-test (*p* < 0.05, FDR corrected). Red, blue, and golden rectangles denote phenylpropanoid and benzenoid, fatty acid derivatives, and terpenes classes of volatiles, respectively. The arrows represent the direction of change: increase (brown) and decrease (green). (**B**) Mean quantities of volatiles detected from tomato leaves of plants (N = 5) of non-inoculated control (NI), inoculated with viruliferous whiteflies (TYLCV), or non-viruliferous whiteflies (NV). Mean quantities were compared by one-way ANOVA Tukey’s HSD test, and significant differences (*p* < 0.05) are indicated by different letters.

**Figure 3 insects-13-00840-f003:**
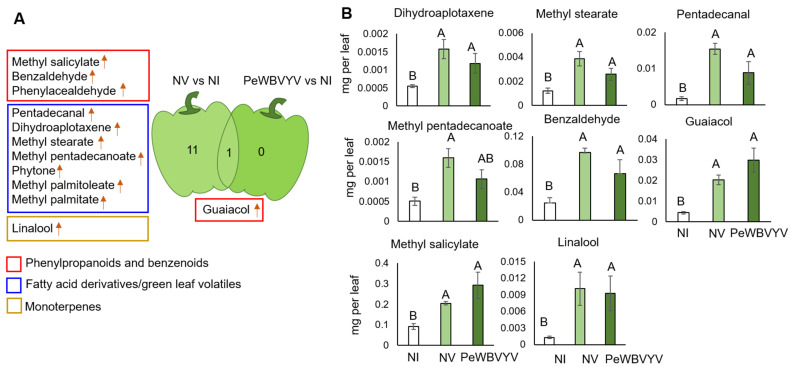
Altered VOC profiles of pepper plants. (**A**) Venn diagram representing the altered VOC profiles of pepper plants inoculated with viruliferous and non-viruliferous whiteflies compared to non-inoculated control plants. Red, blue, and golden rectangles denote phenylpropanoid and benzenoid, fatty acid derivatives, and terpenes class of volatiles, respectively. (**B**) Mean quantities of volatiles detected from pepper leaves of plants (N = 5) of non-inoculated control (NI), inoculated with viruliferous whiteflies (PeWBVYV), or non-viruliferous whiteflies (NV). Mean quantities were compared by Tukey’s HSD test, and significant differences (*p* < 0.05) are indicated by different letters.

**Figure 4 insects-13-00840-f004:**
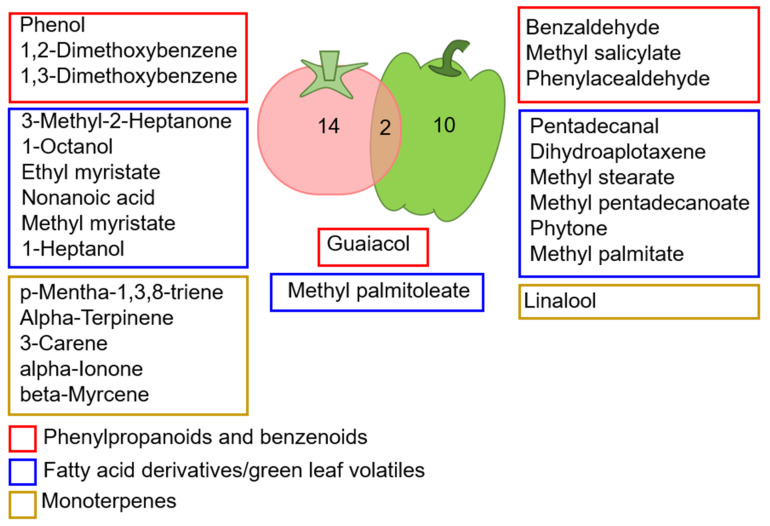
Venn diagrams of all VOCs altered in tomato and pepper plants post inoculation with non-viruliferous/viruliferous whiteflies versus the non-inoculated control.

## Data Availability

Data is contained within the article or Appendix A.

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
