# Peer review of "Comparative Analysis of Volatiles Emitted from Tomato and Pepper Plants in Response to Infection by Two Whitefly-Transmitted Persistent Viruses"

_insects, 2022, doi:10.3390/insects13090840_

Round 1

Reviewer 1 Report

This is an interesting, valuable study that builds on our specific and broad understanding of interactions between plants, an insect vector, and virus. I have no substantial technical issues, but I do have various corrections and suggestions relative to presentation and language.

There are places where the language is insufficiently precise. I will highlight the most important of these that I noted. My general observation is that making certain "casual" sentences are precisely accurate is as important as doing the same for key sentences in the R and D.

l 61 the opening phrase is not logically consistent with the point of the sentence (all sessile organisms don't emit VOCs); I'd just kill the phrase and start "Plants emit..." I assume the actual intent here was to observe that VOCs offer a mechanism for plants to circumvent some of the limitations of being sessile.

l 78 this statement "Bemisia tabaci inflict minimal damage..." is wrong on a couple levels. Using the concepts of injury (action of insect on plant) and damage (response of plant to insect) (see Pedigo et al. 19866 or Higley and Pedig,  1996 for discussion), the cumulative injury determines damage and at sufficiently high levels of injury the damage can be severe. I suspect the intent here was that the injury by individual whiteflies is slight, but even here small injury might release phytohormone or other chemicals in a way different from other injury types. The safe statement is that relative to direct damage from B. tabaci, damage from virus transmission is much, much greater.

l 88 "...studied in the past",  "in the past" is redundant

l 110 "The outcome of this study would be crucial..." Ouch, reads like copied from grant application, and use of "crucial" is way overblown. "Crucial" implies essential, that no progress in future understanding is possible without this study. I think not, and it projects a air of arrogance I'm sure you don't want.

2. M and M

This may be nothing, but whenever I have used, or see used insect or other cages, I worry about light reduction interacting with plant physiological responses. Perhaps this comes from working for years on insects and photosynthesis, but if there is anything quantitative you can say about the light environment within cages, it would be helpful (and reassuring to people like me).

Figure 2. very frustrating. I hate your use of NV, NI, and TYPCV on the Venn diagram, because the abbreviations force the reader to scour the figure description. A much more significant issue is the lack of a legend in the figure for the different color bars. I think adding this is essential.

Fig 3 - comments as Fig 2

l 252 You may disagree, but as there are many plants and many whitelfies, I think thee are many "Interactions" so i'd change this to "Interactions of ... induce changes..."

l 260 "following" typo

The discussion currently consists of two long paragraphs, however these could easily be split into multiple paragraphs based on the range of topics in each. This simple change would significantly enhance readabilty

Author Response

Line 61: Remove plants being sessile

Response: Changed as suggested by the reviewer.

Line 78: Edit B tabaci inflicts minimal damage

Response: The sentence has been rephrased as suggested.

Line 88: In the past is redundant

Response: ‘In the past’ has been deleted

Line 110: Remove ‘crucial’

Response: ‘crucial’ has been altered with ‘helpful’

M&M: Mention light conditions

Response: Photoperiod conditions are mentioned under the conditions of the controlled environment

Fig 2, 3: Abbreviations of NV, NI, TYLCV, PeWBVYV

Response: All abbreviations are expanded in the figure caption.

Line252: ‘many plants and many whitefly’ exist.

Response: Sentence rephrased as suggested.

Reviewer 2 Report

Manuscript ID: insects-1892196

The study aimed at identifying and comparing profiles of VOCs emitted by tomato and pepper plants in response to infection of two whitefly-transmitted viruses, and by whitefly infestation without infection. Significant observations were that plants emitted shared VOCs, but that unique VOC signatures were identified in the different treatments indicating that plants are capable of responding specifically to a particular virus infection as well to an insect infestation without infection. Given the important role of VOCs in plant defence mechanisms, these results have implications for new and improved whitefly management and monitoring programmes.

The manuscript has interesting and novel findings. The aim and knowledge gap are clear; the colour-coded figures are well presented and easy to interpret. It is recommended for publication with minor comments and corrections to the text as follows:

Line 84: remove the extra “and”

Line 104: Remove “Although”

Section 2.1: How was the non-viruliferous status of B. tabaci verified? Perhaps this should be described, or at least the source of these insects cited in the text

Section 2.1: Insects were reared and inoculated under controlled conditions which is standard practice in these types of experiments. However, there could be some mention in the discussion section about how the pathosystems used in the study would compare with field culturing conditions for these crops in terms of VOC emission upon exposure to non-viruliferous and viruliferous insects. Maybe field trials have been conducted, or are planned?

Section 2.2: For PCR confirmation of infection or non-infection status, no details of the PCR assay for TYLCV are given (primers, template, cycling conditions etc). This is also the case for the RT-PCR assays for PeWBVYV infection. Details of the methodology and results of PCR and RT-PCR assays would add value to the manuscript either as citation/s or in supplementary material. It is also not mentioned whether the non-infection status of the pepper plants with aphid-transmitted polerovirus was confirmed prior to treatments

Section 2.3: Plant leaves were used for volatile extraction and analysis. In lines 61-63 of the manuscript, it is stated that plants emit VOCs into the surrounding atmosphere. It may not have been possible to analyse the surrounding atmosphere in the pathosystem set-up, but it would be interesting to include some discussion/speculation about how leaf VOC profiles would differ (or not) from atmospheric profiles

Section 3.2, line 222: Should read “…with non-viruliferous whiteflies.”

Line 260: “following”

Lines 266 and 269: correct spelling of inoculation and inoculated

Line 278: “previously studied…”

Figure 1 legend: “yellowing”

Figure 2 legend: “represent”

Author Response

Line84: Remove ‘and’

Response: Correction has been made.

Line 104: Remove ‘although’

Response: Correction made.

Section 2.1: How was the non-viruliferous status verified?

Response: A sentence has been added regarding the PCR testing along with relevant citations describing the methods.

Section 2.1: How would the VOC profiles differ in field conditions for these crops? may be field trials have been conducted or planned? Mention in the discussion

Response: We added this comment to the discussion.

Section 2.2: Confirmation of infection status of pepper plants and methodology of RT-PCR has not been mentioned.

Response: Citations detailing the methods of PCR/RT-PCR have been included now.

Section 2.3: Mention in discussion how leaf VOC profiles would differ from atmospheric profiles.

Response: We mention it in the discussion

Section 3.2: line 222: ‘non-viruliferous whiteflies’

Response: change made.

Line 260: ‘following’

Response: Correction made.

Line 266,269: spell check inoculation and inoculated.

Response: Change made.

Line 278: Previously studied.

Response: Corrected.

Figure 1 legend: ‘yellowing’

Response: corrected.

Figure 2 legend: ‘represent’

Response: correction made.

Reviewer 3 Report

The manuscript “Comparative analysis of volatiles emitted from tomato and pepper plants in response to infection by two whitefly-transmitted persistent viruses” by Ghosh et al. compares the VOCs profiles of virus infected and non-infected tomato and pepper plants. The results shown here are very interesting and valuable to the research on insect-transmitted plant viruses and the experimental design is adequately described. However, the discussion part is a little poor and needs a more in-depth analysis of the authors’ findings, especially about the compounds (or phytohormones) that were found to be significantly altered among treatments. Please find below my comments on the manuscript:

Line 16. Change “may” to “many” agricultural crops

Line 17. Remove “the” before worldwide agriculture

Line 19. Change “with regards” to “in regards”

Lines 27-30. This sentence is a bit confusing. Please rephrase to make it clearer.

Line 29. B. tabaci should be written in italics

Line 66. Add “a” before “key role”

Line 84. Remove one “and”

Line 116. What kind of cages? Size and material of the cages should be included here

Line 131. I think it would be more suitable to include the growth stage of the plants here.

Line 140. Were the pepper plants 2-weeks old as well? (again I would recommend to refer to the age of the plants as a growth stage and not as a time reference).

Line 147. Change “since” to “of”

Lines 147-148. The same 3 leaves were used for the headspace collection? Please rephrase these 2 sentences to make them clearer.

Line 208. How come n=5? It is mentioned above that 3 leaves were used for the collection of volatiles from each treatment.

Line 224. Change “but” to “except for”

Line 264. Delete “a” and change “profile” to “profiles”

Line 265. Change “belong” to “belonging”

Line 270. Change “sesquterpene” to “sesquiterpene”

Line 278. Change “studies” to “studied”

Author Response

Line 16: Change ‘may’ to many.

Response: correction made.

Line 17: Remove ‘the’ before world-wide.

Response: Correction made.

Line 19: Change ‘with regards’ to ‘in regards’.

Response: Correction made.

Line 27-30: Rephrase.

Response: Sentence has now been rephrased for easier understanding.

Line 66: Add ‘a’ before key role.

Response: Change made.

Line 84: Remove ‘and’

Response: changed.

Line 131: include growth stage of plants.

Response: The leaf stage of plants has now been included.

Line 140: Were the pepper plants 2 weeks old?

Response: The leaf stages of the pepper plants inoculated have now been included.

Line 147: Change ‘since’ to ‘of’

Response: correction made.

Line 147-148: The same 3 leaves used for collection?

Response: This has been corrected as the 3rd, 4th and 5th leaf from the top.

Line 208: how n-5?

Response: N=5 denotes the 5 biological replicates/plants inoculated for each treatment. Three leaves were collected from each biological replicate and mixed to form a single sample.

Line 224: change ‘but’ to ‘except for’

Response: Correction made.

Line 264: Delete ‘a’ and change profile to ‘profiles’

Response: correction made.

Line 265: Change ‘belong’ to ‘belonging’

Response: Correction made.

Line 270: Correct sesquiterpene

Response: correction made.

Line 278: Change ‘studies’ to ‘studied’

Response: Correction made.